# Alphaherpesvirus gB Homologs Are Targeted to Extracellular Vesicles, but They Differentially Affect MHC Class II Molecules

**DOI:** 10.3390/v12040429

**Published:** 2020-04-10

**Authors:** Kinga Grabowska, Magda Wąchalska, Małgorzata Graul, Michał Rychłowski, Krystyna Bieńkowska-Szewczyk, Andrea D. Lipińska

**Affiliations:** Laboratory of Virus Molecular Biology, Intercollegiate Faculty of Biotechnology, University of Gdańsk, Abrahama 58, 80-307 Gdańsk, Poland; kinga.grabowska@biotech.ug.edu.pl (K.G.); magda.wachalska@phdstud.ug.edu.pl (M.W.); malgorzata.graul@biotech.ug.edu.pl (M.G.); michal.rychlowski@biotech.ug.edu.pl (M.R.); krystyna.bienkowska-szewczyk@biotech.ug.edu.pl (K.B.-S.)

**Keywords:** extracellular vesicles, exosomes, alphaherpesvirus, glycoprotein B, MHC (major histocompatibility complex) class II, CD63

## Abstract

Herpesvirus envelope glycoprotein B (gB) is one of the best-documented extracellular vesicle (EVs)-incorporated viral proteins. Regarding the sequence and structure conservation between gB homologs, we asked whether bovine herpesvirus-1 (BoHV-1) and pseudorabies virus (PRV)-encoded gB share the property of herpes simplex-1 (HSV-1) gB to be trafficked to EVs and affect major histocompatibility complex (MHC) class II. Our data highlight some conserved and differential features of the three gBs. We demonstrate that mature, fully processed BoHV-1 and PRV gBs localize to EVs isolated from constructed stable cell lines and EVs-enriched fractions from virus-infected cells. gB also shares the ability to co-localize with CD63 and MHC II in late endosomes. However, we report here a differential effect of the HSV-1, BoHV-1, and PRV glycoprotein on the surface MHC II levels, and MHC II loading to EVs in stable cell lines, which may result from their adverse ability to bind HLA-DR, with PRV gB being the most divergent. BoHV-1 and HSV-1 gB could retard HLA-DR exports to the plasma membrane. Our results confirm that the differential effect of gB on MHC II may require various mechanisms, either dependent on its complex formation or on inducing general alterations to the vesicular transport. EVs from virus-infected cells also contained other viral glycoproteins, like gD or gE, and they were enriched in MHC II. As shown for BoHV-1 gB- or BoHV-1-infected cell-derived vesicles, those EVs could bind anti-virus antibodies in ELISA, which supports the immunoregulatory potential of alphaherpesvirus gB.

## 1. Introduction

Alphaherpesviruses, members of the *Herpesviridae* family, belong to the most widespread human, farm and wild animal pathogens. Herpesviruses have mastered the ability to interfere with the host immune system, which allows them to establish a latent (dormant) infection. Recent studies have unveiled the participation of extracellular vesicles (EVs) shed by cells infected with such herpesviruses as human cytomegalovirus (HCMV), Kaposi’s sarcoma herpesvirus (KSHV), Epstein–Barr virus (EBV) or an alphaherpesvirus—herpes simplex 1 (HSV-1), in the formation of antiviral immunity [1,2,3,4,5]. Those findings have expanded the already impressive collection of known herpesvirus immunomodulatory strategies.

Extracellular vesicles (EVs) represent a heterogeneous population of membranous vesicles released into the extracellular milieu by prokaryotic and eukaryotic cells. EVs differ in their origin, morphology, size, density, and cargo that may be partially specific for an EVs fraction [6,7]. Exosomes represent small EVs of endosomal origin, 30–150 nm in diameter, released by cells as a consequence of intraluminal vesicle (ILV) formation within multivesicular bodies (MVB), and their subsequent fusion with the plasma membrane. Larger EVs may contain plasma membrane-derived microvesicles (50–1000 nm in size) or apoptotic bodies (50–5000 nm in size). EVs can be found in vivo in various body fluids and in vitro in cell culture supernatants. Their cargo may contain proteins, small signaling molecules, and various species of nucleic acids, enlisted in the constantly expanding Vesiclepedia [8,9]. According to the current knowledge, at least some vesicle components are specifically sorted to EVs by certain still extensively studied mechanisms [7,10,11].

Since EVs re-emerged into the scientific world as important mediators of intercellular communication, oncogenesis, immune activation, and many other physiological and pathological processes, their participation in viral pathogenesis has been explored [4,12]. However, the studies on EVs from herpesvirus-infected cells face a hurdle resulting from the similar size of smaller EVs and herpesvirus particles, either representing complete enveloped virions in the range of 140–200 nm or non-infectious light particles (L-particles) reviewed in [13]. Those similarities limit the use of some widely acknowledged techniques of EVs isolation, like size-exclusion chromatography (SEC). The herpesvirus assembly and exosome biogenesis pathways may, at some points, intersect, which was demonstrated in particular for human herpesvirus 6 [14]. Alphaherpesvirus virion morphogenesis and exosome formation share specific components of the endosomal sorting complex required for transport (ESCRT) machinery, such as components of the ESCRT-III complex and Vps4 ATPase [15,16]. Incorporation of herpesvirus material to EVs has been reported, which motivates researchers to test the application of EVs as therapeutics and biomarkers—for example, in liquid biopsies [4,17,18].

Herpesvirus envelope glycoprotein B (gB) is not only an essential component of the virus entry complex but also one of the best-documented (for HCMV and HSV-1) EVs-incorporated viral proteins [1,4,17,19]. During the initial steps of herpesvirus infection, gB, together with other glycoproteins, like gH/gL, forms the core fusion complex [20,21,22]. Whereas gB has been studied in detail as a player in virus entry, less is known about specific roles this glycoprotein plays during later phases of infection. When virus components are produced, gB seems to have a unique property to modify the endosomal–exosomal pathway in a cell and the architecture of early and late endosomes. Expression of HSV-1 gB can affect the trafficking of major histocompatibility complex class II molecules (MHC II) [19,23]. As a result, the surface MHC II levels decrease, and their sorting to exosomes becomes more intensive. In our research on immunomodulatory strategies of alphaherpesviruses, we exploit pathogens of veterinary importance, namely bovine herpesvirus 1 (BoHV-1) and suid herpesvirus 1 or pseudorabies virus (PRV), as research models. gB is the most conserved glycoprotein of herpesviruses with regards to both the amino acid sequence homology and structure [24]. BoHV-1, PRV and HSV-1 gB demonstrate 90%–95% amino acid sequence agreement. In this study, we asked the question of whether incorporation of gB to EVs and the ability to affect MHC class II levels are conserved features that can also be documented for gB homologs of BoHV-1 and PRV.

## 2. Materials and Methods

### 2.1. Cells and Viruses 

Madin–Darby bovine kidney (MDBK) cells (CCL-22, ATCC, Manassas, VA, USA), human melanoma Mel JuSo cells (MJS, a kind gift from Dr. Emmanuel Wiertz, University Medical Center Utrecht, Utrecht, The Netherlands), immortalized porcine alveolar macrophages (PAM, clone 3D4/2, AddexBio, San Diego, CA, USA) and human acute T leukemia cell line Jurkat (E6.1, European Collection of Authenticated Cell Cultures, Salisbury, UK) were cultured in RPMI 1640 (Corning, Corning, NY, USA) supplemented with 10% fetal bovine serum (FBS, Thermo Scientific, Waltham, MA, USA) and Antibiotic Antimycotic Solution (Thermo Scientific, Waltham, MA, USA). Swine kidney SK6 and African green monkey kidney Vero cells (CCL-81, ATCC, Manassas, VA, USA) were cultured in Dulbecco’s modified Eagle’s medium (DMEM, Corning, NY, USA) supplemented as above. GP2–293 cells (Takara/Clontech, Kusatsu, Japan) used for retrovirus production, were cultured in Iscove’s modified Dulbecco’s medium (IMDM, Lonza, Basel, Switzerland), supplemented as above. BoHV-1 field strain Lam (a kind gift from Dr. Frans Rijsewijk, Institute for Animal Health and Science, Lelystad, The Netherlands) was propagated and titrated on MDBK cells. HSV-1 syn 17+ strain (a kind gift from Dr. Walter Fuchs, Friedrich Loeffler Institut, Federal Research Institute for Animal Health, Greifswald-Insel Riems, Germany) was propagated and titrated on Vero cells. PRV strain NIA-3 was propagated and titrated on SK6 cells. SK6 and PRV were kindly provided by Dr. Liesbeth Jacobs, Institute for Animal Health and Science, Lelystad, The Netherlands. The expression of swine leukocyte antigen (SLA) class II was boosted on PAM by 36 h incubation with 0.1 mg mL^−1^ of recombinant swine interferon (IFN)-γ (Bosterbio, Pleasanton, CA, USA) prepared according to the manufacturer’s instructions. For enhanced gB expression, Jurkat cells were activated with 100 nM phorbol 12-myristate 13-acetate (PMA) and 1 mg mL^−1^ phytohemagglutinin (PHA, both reagents from Merck/Sigma, Darmstadt, Germany).

### 2.2. Generation of Stable gB-Expressing Cell Lines

*UL27* (gB gene; gene ID: 24271469 for HSV-1 gB; gene ID:2952558 for PRV gB; genomic sequence JX898220 region: 55564–58365 for BoHV-1 gB) sequences were cloned in retroviral vectors downstream of a retroviral promoter. HSV-1 gB and BoHV-1 gB genes were amplified from isolated viral DNA using WALK polymerase (A&A Biotechnology, Gdynia, Poland) and the following primers: HSV-1 forward 5′-CGGGATCCGCCATGCGCCAGGGC-3′ and reverse 5′-GGAATTCACAGGTCGTCCTCGTCGGCG-3′; BoHV-1 forward 1Fw 5′-CCGTCTTTGCGTCCGTCTTCCAC-3′, 3Fw 5′-CTGGCGCGCTCGAACGGCAC-3′, and reverse 2Rev 5′-CGCCGCGGCGGCGAACAG-3′, 4Rev 5′ GCGCACGTCCCGCCTCCAAG 3′. BoHV-1 gB was isolated in two fragments, joined using the internal XhoI site in the pJET1.2 plasmid for PCR product cloning (Thermo Scientific), and amplified by another PCR reaction with the primers, introducing restriction sites for cloning: Bam Fw 5′-CGGGATCCACCATGGCCGCTCGC-3′, Eco Rev 5′-GGAATTCTCATGCCCCCCCGACGTC-3′. PCR products were verified by DNA sequencing and introduced into BamHI and EcoRI sites of the pLZRS-IRES-Δnerve growth factor receptor (NGFR) vector [25]. The PRV gB-encoding sequence was cloned from the pFB-CMV-VSVG-gB plasmid [26] in the EcoRI site of pLZRS-IRES-ΔNGFR. BoHV-1 gD-encoding sequence (JX898220 region: 118819–121359) was amplified from isolated viral DNA using WALK polymerase and the following primers: forward 5′-GCGGATCCAACATGCAAGGGC-3′ and reverse 5′-GGAATTCCGCTCACCCGGGC-3′. The PCR product was verified by DNA sequencing and introduced into BamHI and EcoRI sites of the pBABEpuro vector [27]. The retroviral packaging system was used to obtain recombinant retroviruses. GP2-293 packaging cells were co-transfected with pCMV-VSV-G (Cell Biolabs, San Diego, CA, USA) and pLZRS-IRES-ΔNGFR-gB or empty pBABEpuro vector (to obtain the control MJSpuro, MDBKpuro, and SK6puro cells) or pBABEpuro-gD. After 24 hrs post-transfection, the medium was refreshed. Virus-containing supernatants were collected after 48 h and used for transduction of enlisted cell lines in the presence of 0.01 mg mL^−1^ polybrene (Merck, Fort Kenner, NJ, USA). The pLZRS-derived cells were stained with anti-NGFR antibodies (Merck/Sigma, St. Louis, MO, USA) and phycoerythrin (PE)-conjugated IgG (Becton Dickinson, Franklin Lakes, NJ, USA), and subsequently sorted using a FACS Calibur flow cytometer Sorting Option (Becton Dickinson, Franklin Lakes, NJ, USA), to obtain cell lines at least 95%-positive for NGFR. MJSpuro, SK6puro, PAMpuro and MJS-gD cells were selected with 2 µg mL^−1^ of puromycin (Merck/Sigma); MDBKpuro required 4 µg mL^−1^ of puromycin for selection.

### 2.3. Antibodies 

Anti-HSV-1/2 gB mouse monoclonal antibody (mAb) (clone 10B7, Santa Cruz Biotechnology, Dallas, TX, USA), mouse anti-BoHV-1 gB mAb (clone H2, VMRD, Pullman, WA, USA), caprine polyclonal anti-IBR/BoHV-1 serum (VMRD), polyclonal PRV gB-specific rabbit serum (a kind gift from Dr. Hanns-Joachim Rziha, Friedrich Loeffer Institut, Federal Research Institute for Animal Health, Tübingen, Germany) or mouse anti-PRV gB mAb (clone A20-c26, also a gift from Dr. Rziha) were used to detect gB. Exosomal markers were visualized with mouse anti-Alix mAb (clone 3A9, Santa Cruz Biotechnology), mouse anti-CD63 mAb (clone MX-49.129.5, Santa Cruz Biotechnology, Dallas, TX, USA), mouse anti-CD9 mAb (clone MCA-469GA, Bio-Rad/AbD Serotec, Hercules, CA, USA) and rabbit anti-flotillin-2 mAb (Santa Cruz Biotechnology). Other cellular markers were detected by goat polyclonal anti-calnexin antibody (Santa Cruz Biotechnology) and rabbit polyclonal anti-Tom40 antibody (Santa Cruz Biotechnology). Mouse anti-HLA-DR mAb (clone L243, Santa Cruz Biotechnology) and mouse anti-bovine MHC class II mAb (clone IL-A21, collected as hybridoma cell culture supernatant, Merck/Sigma) were used in flow cytometry and immunoprecipitation (IP). Rabbit polyclonal anti-HLA-DRα and goat polyclonal anti-CD63 antibodies (Santa Cruz Biotechnology) were used in immunofluorescence. Other antibodies used for immunoblotting were mouse anti-HLA-DRα mAb (clone G-7, Santa Cruz Biotechnology), mouse anti-HLA-DRβ mAb (clone TAL14.1, Santa Cruz Biotechnology), mouse anti- HSV-1/2 glycoprotein C mAb (clone 3G9, Santa Cruz Biotechnology), mouse anti-HSV-1/2 glycoprotein D mAb (clone DL6, Santa Cruz Biotechnology), mouse anti-HSV-1/2 ICP5 mAb (clone 3B6, Santa Cruz Biotechnology), rabbit polyclonal anti-HSV-1 gamma 34.5 protein (ICP34.5) antibody (Bioss Antibodies, Woburn, MA, USA), mouse anti-BoHV-1 glycoprotein C mAb (clone F2, VMRD) and mouse anti-BoHV-1 glycoprotein D mAb (VMRD). BoHV-1 VP5-specific rabbit serum was a generous gift from Dr. S. van Drunen Littel-van den Hurk (University of Saskatchewan, Saskatoon, SK, Canada). Mouse anti-BoHV-1 glycoprotein E mAb was a kind gift from Dr. Frans Rijsewijk, Institute for Animal Health and Science, Lelystad, The Netherlands. Rabbit polyclonal antibodies specific to BoHV-1 glycoprotein M and the UL49.5 protein were previously described [25]. PRV gC-specific mouse antibody (B16-d6), polyclonal PRV gD-specific rabbit antibody (016/00), and mouse anti-PRV glycoprotein E mAb were kindly provided by Dr. Hanns-Joachim Rziha. The antibodies used for immunofluorescence were rabbit polyclonal anti-HLA-DRα (Santa Cruz Biotechnology); markers of early (EEA1), recycling (Rab11), or late (Rab7) endosomes and endolysosomal organelles (LAMP1) were stained with the Endosomal Marker Antibody Sampler Kit (Cell Signaling Technology, Danvers, MA, USA). Goat anti-mouse horseradish peroxidase (HRP) IgG, donkey anti-rabbit HRP-conjugated IgG, and donkey anti-goat HRP-conjugated IgG (Jackson Immunoresearch, West Grove, PA, USA) were used as secondary antibodies in immunoblotting and ELISA.

### 2.4. Extracellular Vesicles Isolation

EVs were isolated from the supernatants of cells grown in serum-free medium (Hybridomed DIF-1000, supplemented as described in [19]; Biochrom, Berlin, Germany) for 48 h to avoid contamination with serum-derived products. The average culture size was three times the size of a culture dish with a 150 mm diameter. The initial confluence of cell culture (at the moment of medium replacement) was 70%. The supernatants were used for size-exclusion chromatography (SEC) isolation of EVs according to the protocol described in [28], with minor modifications. To remove cellular debris, the supernatants were precleared by centrifugation for 10 min at 300× *g*, 20 min at 2000× *g*, and 30 min at 10,000× *g*. Subsequently, the supernatants were filtered with 0.45 μm filters with the polyvinylidene fluoride membrane, to remove larger vesicles and concentrated by ultrafiltration using Amicon^®^ Ultra-15 Centrifugal Units 100-K (Merck) to reduce the volume to 1 mL. The SEC column (10 mL plastic syringe, Becton Dickinson) was stacked with Sepharose CL-4B (GE Healthcare, Chicago, IL, USA) in PBS containing 0,32% (*w*/*v*) trisodium citrate, pH 7.4 (0.22 µm-filtered), to create a 16 mm × 62 mm column. The concentrated supernatant was loaded on the column, followed by elution with PBS/citrate, to collect twenty-five sequential fractions of 0.5 mL, subsequently analyzed by immunoblotting and transmission electron microscopy or stored at −80 °C for further studies.

For isolation of EVs from virus-infected cell culture supernatants, we followed the procedure established by [4]. The cells were seeded for infection in the complete medium in 150 mm culture dishes. For infection, the cells were washed three times with PBS and the virus inoculum was applied in the serum-free medium (Hybridomed). MJS cells were infected with HSV-1 at a multiplicity of infection (moi) of 0.1 or with BoHV-1/PRV at a moi of 0.5. MDBK cells were infected with BoHV-1 at a moi of 0.1. After 2 h of incubation, the inoculum was aspirated, the cells were washed with the serum-free medium and covered with the fresh serum-free medium for 48 h. Infected or mock-infected cell supernatants were collected, precleared, filtered, and concentrated to 0.5 mL as described above. The supernatant was diluted with 60% (*w*/*v*) iodixanol (OptiPrep, Progen Biotechnik, Heidelberg, Germany) to obtain 6%, and loaded on top of ioxidanol-based gradient ranging from 6 to 18%, with a 1.2% increment (11 mL in volume). The fractions were prepared by dilution of 60% iodixanol in 10 mM Tris-HCl, pH 8.0, and 0.25 M sucrose. Samples were ultracentrifuged in an SW41 Ti rotor for 5 h at 200,000 g, 4 °C in a Beckman Coulter Optima L-90K ultracentrifuge (Beckman Coulter, Brea, CA, USA). Twenty-two fractions (0.5 mL, with the top fraction) were collected from top to bottom and analyzed. The number of infectious virus particles in each fraction was determined by a plaque assay on Vero (HSV-1), MDBK (BoHV-1) or SK6 (PRV) cells, as described in [29].

### 2.5. Immunoblotting and Immunoprecipitation

For immunoblotting, the cells were lysed in the Cell Lytic M buffer (Merck/Sigma); for immunoprecipitation, mild lysis buffer (1% (*w*/*v*) 3-((3-cholamidopropyl) dimethylammonio)-1-propanesulfonate (CHAPS) in 10 mM Tris-HCl, pH 7.5, 150 mM NaCl, 0.5 mM EDTA, 1 mM CaCl_2_) was applied. EVs were lysed in 5× RIPA lysis buffer (150 mM NaCl, 1% Nonidet P-40, 50 mM Tris-HCl, pH 7.6, 0.1% sodium dodecyl sulfate, 5 mM EDTA, 0.5% sodium deoxycholate). The buffers were supplemented with the cOmplete mini protease inhibitor cocktail (Roche, Basel, Switzerland). Total protein concentration was estimated by the BCA Protein Assay (Santa Cruz Biotechnology) or measured at 280 nm using DS-11 Spectrophotometer (DeNovix, Wilmington, DE, USA). Cell or EVs lysates (3 μg total protein or otherwise stated) were analyzed by SDS-PAGE and immunoblotting directly or incubated with anti-gB or anti-HLA-DRα molecules together with Protein-A Sepharose beads (Merck/Sigma) to isolate protein complexes. For analysis of N-linked glycosylation, equivalent amounts of cell lysates were incubated with endo-β-N-acetylglucosaminidase H (EndoH; New England Biolabs, Ipswich, MA, USA) or peptide:N-glycosidase F (PNGaseF; New England Biolabs, Ipswich, MA, USA) according to the manufacturer’s conditions. Cell lysates and immunoprecipitated proteins were separated in SDS-PAGE and immunoblotted as described before [25]. CD63 and CD9 were analyzed in non-reducing conditions, as it is recommended for their detection [30]. The optical density of protein bands was measured with the Alliance-1D software of the imaging system (UVITEC, Cambridge, UK).

### 2.6. Immunofluorescence

Cells were grown on microcover glass for 24 hrs. Next, they were fixed with 4% paraformaldehyde in PBS and permeabilized with 0.2% (*w*/*v*) Triton X-100 in phosphate-buffered saline (PBS) for 8 min. After washing with PBS, cells were incubated with mouse anti-gB antibodies (1:1000) in combination with the rabbit anti-DRα chain of MHC class II molecules (1:1000) and goat anti-CD63 (1:1000) for 1h. After washing, cells were incubated with suitable secondary antibodies. gB was visualized using Alexa 546-conjugated goat anti-mouse IgG (1:1000, Thermo Scientific, Waltham, MA. USA), for MHC class II molecules staining, Alexa 633-conjugated anti-rabbit IgG (1:1000, Thermo Scientific) was applied; CD63 was visualized using Alexa 488-conjugated donkey anti-goat IgG (1:1000, Thermo Scientific). The stained cells were analyzed using a Leica TCS SP8X confocal laser scanning microscope (Leica Microsystems, Wetzlar, Germany). The degree of co-localization was analyzed by the Leica Application Suite X software and quantified using Pearson’s correlation coefficient.

### 2.7. Flow Cytometry, Internalization and Export Assays

Cell surface expression of specific molecules was determined by indirect immunofluorescence using primary antibodies as indicated and, as a second step, phycoerythrin (PE)-conjugated goat anti-mouse IgG (1:350, Becton Dickinson). Cells were stained as above or with secondary antibodies only as controls. For double staining of unsorted gB-retrovirus-transduced cells, mouse FITC-conjugated anti-NGFR mAb in combination with mouse allophycocyanin (APC)-conjugated anti-MHC II (L243) mAb was applied for 1 h on ice.

The kinetics of MHC II internalization and the appearance at the surface of MJSpuro (as a positive control), and HSV-1 or BoHV-1 gB-expressing MJS cells were assessed as described in [31], with slight modifications. Surface MHC II was stained with saturating amounts of L243 antibody in the culture medium with 2% (*w*/*v*) FBS for 1 h on ice. Then the cells were washed three times with the medium-2% FBS, aliquoted in the medium-2% FBS, and kept on ice (time point 0) or shifted to 37 °C for 20 or 40 min. After incubation, cells were cooled on ice to inhibit further vesicular transport. The remaining MHC II was detected by staining with PE-conjugated goat anti-mouse IgG for 45 min on ice. Cells were analyzed by flow cytometry.

To establish the kinetics of MHC II appearance at the surface, MJSpuro or HSV-1/BoHV-1 gB MJS were incubated with the saturating amounts of L243 antibody in the culture medium with 2% (*w*/*v*) FBS for 1 h on ice, washed three times with the cold medium-2% FBS and placed in the fresh medium−2% FBS with the saturating amounts of APC-conjugated L243 antibody that should bind only to the new MHC II molecules arriving at the surface. Cells were next aliquoted, kept on ice (time point 0), or shifted to 37 °C for 20 or 40 min. After cooling on ice and washing, the cells were analyzed by flow cytometry.

Cells were analyzed using a FACS Calibur flow cytometer (Becton Dickinson) and the CellQuest Pro software or (for MHC II internalization and export assays) using Merck/Guava easyCyte flow cytometer (Merck) and the InCyte software.

### 2.8. Transmission Electron Microscopy (TEM)

For visualization of the particles, 10 µL droplet of the EVs sample or the iodixanol gradient fraction (diluted 1:5 in PBS) was deposited on formvar–carbon-coated 200-mesh nickel grids. Negative staining was performed using 2% uranyl acetate. Following the staining, grids were studied using transmission electron microscope Tecnai G2 Spirit BioTWIN (FEI Inc., Hillsboro, OR, USA) at 120 kV, at the Laboratory of Electron Microscopy, University of Gdańsk or JEM 1400 (JEOL, Akishima, Japan) at 80 kV, at the Laboratory of Electron Microscopy, Nencki Institute of Experimental Biology of the Polish Academy of Sciences (Warsaw, Poland). The diameters of EVs in TEM images from stable cell lines were measured using the Tecnai software (200 vesicles were analyzed).

### 2.9. gB ELISA

ELISA plates with 96 wells (high protein binding, Corning) were coated with cell lysates or EVs samples diluted in PBS (25 μg total protein) for 16 h at 4 °C. Each well was then blocked with 1% (*w*/*v*) BSA in PBS with 0.05% (*v*/*v*) Tween-20 (PBST) for 24 h at 4 °C. Polyclonal caprine antiserum IBR/BoHV-1 (VMRD) was diluted in PBST with 0.1% (*w*/*v*) BSA (1:9000) and added to the wells for 16 h at 4 °C. After the washing, primary antibodies were detected with the anti-goat HRP-conjugated antibodies (1h incubation) and hydrogen peroxide-tetramethylbenzidine (TMB) as a colorimetric substrate. The reaction with TMB was stopped with 0.5 M sulfuric acid, and signal intensity at 450 nm was measured using the Infinite^®^ 200PRO NanoQuant plate reader (Tecan, Männedorf, Switzerland).

### 2.10. Sequence Alignment

For homology alignments of amino acid sequences, the M-coffee tool (http://www.tcoffee.org/Projects/mcoffee) was applied.

## 3. Results

### 3.1. Localization of BoHV-1, PRV, and HSV-1 gB to EVs Is Conserved

#### 3.1.1. Construction of Stable Cell Lines Expressing BoHV-1, PRV or HSV-1 gB, and Isolation of EVs by Size Exclusion Chromatography (SEC)

Alphaherpesviruses can be characterized by relatively short life cycles and can cause lytic infections, which imposes technical limitations on EVs isolation regarding the time of collection and separation of EVs from lysed cellular material. Hence, first, we analyzed the potential of gB homologs to be trafficked via endosomal–exosomal compartments to EVs and interact with MHC II in cell lines that constitutively express gB homologs as single virus components. BoHV-1, PRV, and, for comparison, also HSV-1 gB-encoding sequences were introduced into various cell types using retroviruses. The cells were subsequently sorted to a high purity, based on the truncated nerve growth factor receptor (NGFR) marker expression, to obtain stable gB-expressing cell lines (Figure 1A). The human melanoma MJS cell line produces endogenous MHC II and, for this reason, it has been widely used in MHC II research [32,33,34]. MJS cells are also permissive for BoHV-1 [35] and PRV (our unpublished data). Furthermore, we took advantage of the fact that bovine ATCC-derived MDBK cells constitutively express bovine MHC class II molecules [36]. Human Jurkat T cells and swine kidney SK6 cells were included as MHC II-negative to compare the effect of MHC II expression on gB incorporation in EVs. We also constructed cell lines transduced with puromycin N-acetyl-transferase-expressing retroviruses (MJSpuro, MDBKpuro, and SK6puro), which we believe represent more proper controls for comparing the untransduced parental MJS, MDBK, or SK6 cells. For technical reasons, only in the case of Jurkat were untransduced cells chosen for comparison.

The method used for isolation of vesicles is one of the critical factors that may affect the composition of EVs cargo [37]. We chose size exclusion chromatography (SEC), widely used in EVs research [28,38], to purify the vesicles from stable cell line supernatants. Larger vesicles were excluded in this procedure by ultrafiltration with 0.45 µM filters. Twenty-five SEC fractions were analyzed for the presence of EVs markers by immunoblotting (CD63 for human cells; Alix for MDBK and SK6, as the bovine and swine tetraspanin was undetectable by the commercially available antibodies that we tested). The distribution of the markers for all the cell culture supernatants was typical for SEC-purified vesicles, e.g., from human plasma [28], and ranged from fractions 9 to 13, with a peak fraction of 10 or 11. The observed distribution indicated an effective EVs separation. Samples from the peak fraction were used for further characterization by immunoblotting for the presence of two EVs-associated markers (Alix-95 kDa, flotillin-2–49 kDa), which were clearly enriched in the EVs fraction (Figure 1B–E). The lower molecular weight protein band, detected by the anti-human Alix antibody in bovine and swine cell lysates, may represent unspecific staining. It is noteworthy that we could detect lower amounts of Alix in EVs from HSV-1 gB-expressing MJS and MDBK cells, compared to flotillin, whose targeting of EVs seemed to be unaffected. ER-resident calnexin (with an observed molecular size in SDS-PAGE gels of 90 kDa) and mitochondria outer membrane-associated Tom40 (40 kDa) proteins were undetectable in the EVs samples, which confirms their purity. The EVs size and morphology were finally analyzed by transmission electron microscopy (TEM), depicting typical cup-shaped vesicular structures, ranging from 40 to 180 nm in size (with an average diameter of 81 nm), for all the tested cell supernatants (Figure 1F and Appendix A).

#### 3.1.2. Mature HSV-1, BoHV-1 and PRV gB Localize to Extracellular Vesicles

During HSV-1 infection, gB strongly associates with the membranes of endosomal multivesicular bodies (MVB), the compartment of exosome formation [39]. The incorporation of HSV-1 gB to extracellular vesicles in plasmid-transfected human cells has also been well-documented [19,23]. To determine whether EVs targeting is conserved for BoHV-1 and PRV gB, we analyzed our EVs preparations for the presence of the viral glycoprotein. gB is a type I membrane glycoprotein, composed of 933/913 amino acid (aa) residues for BoHV-1 and PRV, respectively. BoHV-1 and the PRV gB precursor (named gBa, with an apparent molecular weight of 130/120 kDa in reducing SDS-PAGE gels) are cleaved at the position of 502/504 by the cellular furin protease into two subunits, which remain linked by disulfide bonding—the N-terminal gBb (74/68 kDa) and the C-terminal-transmembrane gBc subunit (55 kD) [40,41,42]. We chose to use polyclonal anti-gB sera, which were able to recognize all three gB forms. We could easily detect two processed BoHV-1 and PRV gB fragments enriched in EVs from porcine, bovine, and human cells, whereas the precursor gBa was undetectable (Figure 2A,B). This suggested that EVs-incorporated gB were derived from trans-Golgi (TGN)/post-Golgi compartments, where gB processing occurs. The observed weight of gB species differed slightly between the cell lines, which may have depended on the glycosylation pattern, as gB is both N-and O-glycosylated at several positions [43]. HSV-1 gB (904 aa) lacks a furin recognition site [44]. The 110–120 kDa protein could be detected in both cell lysates and the EVs fractions, although its content in the vesicles from MHC II-negative Jurkat cells was low compared to the cell lysates (Figure 2C). HSV-1 gB was present in cell lysates and MJS-derived EVs as a double band, with several lower molecular weight bands recognized by gB-specific antibodies. Depending on the cell type, gB can be produced in two high-molecular-weight forms, which may represent the fully processed glycoprotein and its precursor, with mixed type N-glycans [44]. Lower molecular weight fragments are rarely reported in plasmid-transfected cells, e.g., in mouse melanoma [45].

N-glycosylation can be a good indicator of glycoprotein maturation status, as mature proteins contain processed N-glycans, which are resistant to endoglycosidase H (endoH). Such N-glycans are also indicative of a trans-Golgi origin. Therefore, we proceeded to analyze the N-glycosylation of the gB in EVs and cell lysates using endoH and peptide:N-glycosidase F (PNGaseF), which removes all N-glycans. All three gB homologs contain six generic motifs for N-glycosylation (recognized by the Eukaryotic Linear Motif resource at http://elm.eu.org). They are distributed evenly in the gBb and gBc of PRV and at a ratio of 4:2 in BoHV-1 gB. According to our immunoblotting analysis, the BoHV-1 gBb subunit contained endoH-resistant modifications, both in cell lysates and EVs (Appendix A). The trimming of high-mannose sugars also seemed to significantly improve the recognition of this gB form by BoHV-1-specific goat serum antibodies. We could detect BoHV-1 gBc or PRV gBb and gBc forms partially resistant to endoH, as endoglycosidase treatment yielded proteins with a molecular weight that was still higher than that of the PNGaseF-deglycosylated polypeptides. These forms, with mixed type sugars, were dominant in EVs, supporting the conclusion that gB passed the secretory pathway at least to TGN, before it was incorporated into EVs. For HSV-1 gB, even the most slowly migrating form contained mixed N-glycans both in cell lysates and EVs, which may indicate that in MJS cells, gB preserves some high-mannose sugars. It is interesting that lower molecular weight HSV-1 gB fragments could also be modified by endoH, with the 50 kDa gB form representing the glycoprotein fragment with partially complex glycans.

### 3.2. Alphaherpesvirus gB Homologs Affect MHC Class II-CD63 Trafficking

In plasmid-transfected MJS cells, HSV-1 gB co-localizes with MHC class II molecules (or human leukocyte antigen) HLA-DR and CD63 [19]. The CD63 tetraspanin is abundantly present in late endosomes and lysosomes and is enriched on the intraluminal endosomal vesicles, which are secreted as exosomes. An enlargement of gB/CD63/DR-costained vesicles was observed in some cell types, usually upon the overexpression of gB [19,23]. In addition, CD63 stably interacts with MHC II at MVB, including ILV, and controls MHC II expression [46]. Therefore, the CD63-MHC II association may be important for exosomal MHC secretion. This encouraged us to investigate the subcellular localization of gB, CD63, and MHC II (through its HLA-DRα chain) in our stable MJS cell lines using immunofluorescence and confocal laser-scanning microscopy (Figure 3). We could find all three gB homologs co-staining with DRα and CD63 in vesicular structures. Besides, the measurement of the Pearson’s correlation coefficient (Table 1) indicated a slightly more intensive CD63 and DR co-localization in the presence of gB. A threshold of the coefficient equal to 0.5 is regarded as a significant co-localization, and in the presence of HSV-1, BoHV-1 and PRV gB, the coefficient increased, on average, from 0.64 to 0.71, 0.76, and 0.74, respectively. We did not observe a significant enlargement of CD63-positive structures, which resembles low-level HSV-1 gB-expressing MJS cells, described by Temme et al. Only the BoHV-1 gB could be detected in large amounts and also perinuclearly in a compartment resembling Golgi.

Next, we tested if gB can also alter the trafficking of other endosomal markers, like the early endosome-associated protein 1 (EEA1), recycling endosome-decorating Rab11, the late endosome marker Rab7, and the endolysosomal LAMP1 (lysosomal-associated membrane protein 1). We could observe a low co-localization of gB with Rab7 and LAMP1 and some co-localization (yellow signal) with EEA1 and Rab11, with the Pearson’s coefficient below the threshold (Appendix A). Nevertheless, all the tested endosomal markers had a similar localization pattern, regardless of the gB expression. It is noteworthy, in this particular analysis, that a small number of HSV-1 gB-MJS contained enlarged gB-positive vesicles, which did not co-localize with EEA1 and Rab11. However, they seemed to co-stain with LAMP1 and Rab7. Possible explanations of this phenomenon could be that, occasionally, high gB-expressing cells could be found with the “giant” late endosome phenotype or that the visualization of EEA1, Rab7, Rab11, and LAMP1 required taking confocal images at depths different than those presented in Figure 3 and more suitable for observations of enlarged vesicles.

### 3.3. Alphaherpesvirus gB Homologs Differentially Affect the Surface Expression of MHC II

HSV-1 gB was shown to alter HLA-DR trafficking to the plasma membrane, which resulted in a decreased expression of the cell surface MHC II [19]. The co-localization of BoHV-1 and PRV gB with HLA-DR caused us to assess the MHC II levels in our stable cell lines. Firstly, we measured the HLA-DR levels by flow cytometry in unsorted retrovirus-transduced cells, which are a mixture of NGFR-gB-positive and gB-negative cells (Figure 4A). This way, we could demonstrate that HSV-1 gB could strongly downregulate HLA-DR. In BoHV-1 gB-positive cells, a reduction in MHC II could also be found, albeit less prominently. PRV gB seemed to have no effect on HLA-DR. Next, we analyzed the mean fluorescence intensities of MHC II in sorted MJS cell lines to obtain more quantitative results (Figure 4B). As an additional control, we also measured the surface MHC II on MJS cells expressing another BoHV-1 envelope protein: glycoprotein D (gD). In the stable cell lines, HSV-1 and BoHV-1 gB shared the property of reducing MHC II (by 60% for HSV-1 and 40% for BoHV-1 glycoprotein), whereas PRV gB seemed to slightly stabilize HLA-DR. On the other hand, the gB expression had no significant effect on the surface CD63, despite the observed co-localization of gB-CD63-DR. What is more, gBs had a similar differential effect on bovine MHC II in stable MDBK cell lines (Figure 4C). While HSV-1 gB could affect bovine MHC II most strongly, BoHV-1 gB had an intermediate effect, whereas, interestingly, PRV gB induced a dramatic increase in the surface bovine MHC II.

These results could suggest a unique specialization of PRV gB towards porcine MHC II (swine leukocyte antigen class II- SLA II). Unfortunately, immortalized porcine cell lines with a high endogenous MHC II expression were unavailable. Immortalized porcine alveolar macrophage cells (PAM) were established [47], and one of the clones was reported as SLA II-positive [48]. We used another PAM clone established by Weingartl et al., 3D4/2. However, we could detect MHC II (SLA-DR) in approximately 3% of cells (Appendix A). This level was too low to ensure a reliable analysis. Hence, we chose to boost MHC II expression using swine interferon γ (IFN-γ), optimizing the conditions to minimize its cytotoxic effect. This way we could reach 40%–55% of BoHV-1 and PRV gB, or puromycin N-acetyl-transferase-expressing PAM, with a detectable SLA-DR. gB homologs or the puro resistance gene were introduced into PAM in a similar way as in the other cell lines. Despite the confirmed expression of gB in PAM (Appendix A), none of the homologs could significantly alter the surface levels of SLA II upon IFN γ treatment (Figure 4D).

### 3.4. Alphaherpesvirus gB Homologs Differentially Interact with MHC Class II Molecules and Affect the Incorporation of HLA-DR to EVs

The mechanism of the HSV-1 gB-mediated downregulation of HLA-DR was attributed to the direct interaction of the viral protein with MHC via a short amino acid motif homologous to the sequence of the human invariant chain (Ii, CD74), consisting of a MHC II groove-binding segment and a promiscuous binding site (PBS) (Figure 5A). During MHC II biosynthesis, Ii secures the peptide-binding groove of MHC II, before the complex can reach the late endosomal MHC II loading compartment (MIIC). This Ii-like sequence is, however, located in one of the least conserved regions of gB, and it seems to differ significantly in BoHV-1 and PRV gB. In BoHV-1 gB, the position of proline residues in the PBS is preserved, but there are no lysine residues, which are thought to mediate the interaction with HLA-DR. This region is, however, followed by a potential MHC II ligand, which could accommodate the groove, with a score of 26, as estimated by the SYFPEITHI database. In PRV gB, the conservation within this region is so low that several candidate sequences could be aligned with the PKPPKP region. They are also followed by strong potential MHC II peptides, yet not in direct proximity.

We sought to investigate, by co-immunoprecipitation (co-IP), whether this lack of Ii resemblance can determine the ability of gB to bind MHC II. We could detect HSV-1 gB-MHC II interaction in MJS and MDBK cell lysates. Immunoprecipitated BoHV-1 gB-HLA-DR or bovine MHC II complexes were also detectable, but in lower amounts that were more challenging to visualize (Figure 5B,C). It is interesting that we could detect high-molecular-weight proteins with gB-specific antibodies in the complexes, which co-migrated with the gB precursor gBa. We could not convincingly detect PRV gB interacting with HLA-DR; only a faint band that might represent DRα was present in the complexes precipitated with gB-specific antibodies (Figure 5D).

HSV-1 gB was reported to divert HLA-DR to exosomes, increasing MHC II levels (demonstrated as DRα by immunoblotting) in the vesicles [19]. Therefore, we analyzed the HLA-DR in MJS gB-derived EVs (Figure 5E). In our stable cell line supernatants, HSV-1 gB could indeed increase HLA-DRα levels. Interestingly, this effect was not so prominent for the HLA-DRβ chain. In the case of BoHV-1 gB-expressing cells, we could demonstrate that the levels of DRα and β were similar to the control MJSpuro cells. The most apparent difference could be denoted for the PRV gB-expressing cells, where significantly lower amounts of HLA-DR could be detected in SEC-isolated EVs. Taken together, our results suggest that HSV-1, BoHV-1 and PRV gB differentially affect MHC II levels, both in cells and cell-secreted EVs.

### 3.5. HSV-1 gB and BoHV-1 gB Affect HLA-DR Export to the Plasma Membrane

It has been postulated that HLA-DR downregulation in the presence of HSV-1 gB results from the interrupted transport route of MHC II complexes to the plasma membrane, rather than from their accelerated internalization by endocytosis [19]. Therefore, using a flow cytometry-based assay, we next assessed the kinetics of HLA-DR internalization and its appearance at the surface of MJS BoHV-1 gB cells and compared these processes with MJS puro or MJS HSV-1 gB cells (Figure 6). In both gB-expressing cell lines, we found a very similar reduction in MHC II within 20 or 40 min at 37 °C, which was statistically comparable to the control cells. These data confirm that both gB orthologs do not significantly alter the rate of MHC II internalization.

When we used a similar flow cytometry-based approach to measure the kinetics of HLA-DR export to the cell surface, a retarded appearance of MHC II could be observed on both gB-expressing cell lines within 20 or 40 min of incubation at 37 °C. The effect of BoHV-1 gB seemed to be, in this respect, even stronger than in the case of its HSV-1 counterpart. In comparison with the MJSpuro cells, HLA-DR could reach levels, on average, 22.3% or 27.7% lower at 20 min and 23.6% or 52% at 40 min (on MJS HSV-1 gB and BoHV-1 gB, respectively). These results indicate that both gB proteins interfere with the HLA-DR export (either from the direct synthesis or recycling, as our assay does not discriminate between these two routes), which can, at least partially, explain the reduced surface MHC II expression.

### 3.6. Alpherpesvirus gB Localizes to EVs Released during HSV-1, BoHV-1 and PRV Infection

With a view to determining the biological significance of gB incorporation in the EVs in our stable cell lines, we tested whether gB can also be transferred to the vesicles during viral infection. An efficient and reliable separation of EVs from virus particles can, however, be challenging, as vesicles and virions demonstrate similar biophysical properties, such as size and sedimentation [49]. We chose to follow a recently reported EVs separation protocol involving ultracentrifugation in a slowly descending discontinuous iodixanol (OptiPrep)/sucrose gradient, developed for HSV-1 by Deschamps and Kalamvoki [4]. As EVs need some time (usually 24–72 h) to accumulate, we infected MJS cells with HSV-1, BoHV-1 or PRV or MDBK with BoHV-1 at low multiplicities of infection (moi): 0.1 for HSV-1 in MJS and BoHV-1 in MDBK or 0.5 for BoHV-1 and PRV in MJS. The kinetics of BoHV-1 and PRV infection in MJS are slower than in species-specific cells, although the progeny virus can be produced ([35] and our unpublished data [50]). The infected cell culture supernatants were collected 50 h post-infection; cell debris, nuclei and larger vesicles were separated by filtration through 0.45-μm-pore size filters. Uninfected MJS/MDBK cell culture supernatants were analyzed as controls. First, equal volume samples (40 µL) of twenty-two gradient fractions (500 μL, with a 1.5 mL top fraction) were analyzed by immunoblotting for the presence of CD63, CD9 or Alix (for MDBK) and the viral glycoproteins C (gC)), D (gD, for HSV-1), E (gE, for BoHV-1 and PRV), and gB (Figure 7 and Appendix A). Infectious viral particles were measured in the fractions by plaque assays (Figure 7).

The results demonstrate that the markers of EVs were concentrated in light-density fractions, usually with numbers 8–12 (counting from the top of the gradient) of the gradients from HSV-1, BoHV-1, and PRV-infected MJS. A similar pattern was characteristic for uninfected MJS and MDBK cells (Appendix A). The OptiPrep densities of these fractions corresponded well to those reported for gradient-purified melanoma vesicles [51]. In supernatants of BoHV-1-infected MDBK cells, Alix was accumulated in fractions 9–13 and then in higher density fractions, with the numbers 17–20. The level of gC and gE in the light-density fractions was low and much lower than in high-density fractions or the bottom fraction, which should represent virions, where the gC/gE signal was evident. On the contary, the three gBs, and also HSV-1 gD, could be easily detected in the light-density fractions. The glycoproteins were present in several fractions that might represent vesicles of different size and density. In the bottom fraction of some gradients, low amounts of EVs markers were also present, especially CD9. We could detect residual virus infectivity in the light-density fractions (up to 2 × 10^5^ for HSV-1 in MJS and 1.2 × 10^4^ for BoHV-1 in MDBK; for BoHV-1 and PRV in MJS, the amount of contaminating infectious particles was marginal), which allowed us to conclude that we could obtain EVs-enriched fractions (number 8–15) and virus-enriched fractions (number 20–22).

Next, we chose fractions with peak EVs marker expressions: F11 (which most probably represents smaller EVs) and F15 (which might represent larger EVs), together with fraction number F21 (virus-enriched) for TEM analysis (Appendix A). In F11 and F15, we could only visualize vesicles characteristic for EVs. F21 contained predominantly membranous structures with an electron-dense capsid-like material, resembling herpesvirus particles [4], mixed with some empty, cup-shaped vesicles. For HSV-1-infected MJS cells, some virions had unusual, partially tubular shapes.

We analyzed the protein content of the chosen fractions more thoroughly using a collection of virus protein-specific antibodies available in our laboratory. We checked, by immunoblotting, whether F11, F15, and F21 (for HSV-1, we analyzed both bottom fractions F21-F22) contained gD (for BoHV-1 and PRV), BoHV-1 gM, capsid protein VP5/ICP5 (for HSV-1 and BoHV-1), HSV-1 neurovirulence factor and protein kinase R inhibitor γ34.5, and BoHV-1 immunomodulatory protein UL49.5 (Figure 7E). Fraction F7 (which may contain protein aggregates and small vesicles) was used as a control. In addition, we blotted the infected MJS samples for HLA-DRβ, as MHC II molecules are also EVs markers. Cell lysates and virus-infected cell lysates were analyzed as controls. BoHV-1 and PRV gD was present in fractions F11–F15, resembling, in this respect, HSV-1 gD. Low levels of BoHV-1 gM and UL49.5 were also localized in the EVs fractions from the infected MDBK gradient, but we could not detect HSV-1 γ34.5 or the HSV-1/BoHV-1 capsid protein there. The amount of VP5 in the BoHV-1 material was generally low, which indicates that low amounts of virus particles were released and also makes the detection of other viral proteins in EVs more reliable (those not resulting from contamination with virions). In the case of DRβ, it was accumulated in the EVs fractions, in much lower amounts in the HSV-1 virion fraction, and it was undetectable in BoHV-1 and PRV-infected MJS-derived F21.

To compare the MHC II incorporation in EVs during alphaherpesvirus infection, we decided to blot F11 fractions from uninfected or virus-infected MJS supernatants for DRβ and normalize them to the flotillin-2 marker (which is more constant than, for instance, Alix) from the same samples. The results shown in Figure 7F indicate that all three alphaherpesviruses were able to increase DRβ incorporation in EVs. The effect of the whole PRV virus was, in this respect, the most evident and contrary to PRV gB-expressing stable cells (Figure 4E).

### 3.7. BoHV-1 gB Transferred by EVs can Bind Virus-Specific Antibodies from Animal Serum In Vitro

To investigate a potential functional consequence of gB incorporation in EVs, we tested whether gB-positive vesicles can bind virus-specific antibodies in ELISA. A commercially available whole BoHV-1 virus-immunized polyclonal serum was used for the detection of BoHV-1 gB in SEC-isolated EVs (unlysed) from stable MJS or MDBK cell culture supernatants (Figure 8A) or from purified OptiPrep fractions, designated here as vF7, vF11 and vF15 (EVs-enriched) or v21 (virus-enriched), from BoHV-1-infected MDBK (Figure 8B) or MJS cells (Figure 8C). EVs from the control MJS/MDBKpuro cells or corresponding OptiPrep fractions from uninfected cells, designated here as ØF11 and ØF21, were used to establish the specificity of binding.

gB-harboring EVs could be efficiently recognized by anti-BoHV-1 serum. MJS gB-released vesicles produced a much stronger ELISA signal than vesicles from MDBK gB. As we used samples with an equal protein content to coat the plates, and the initial analysis of EVs revealed a similar gB content in MJS- or MDBK-derived EVs (Figure 1C), this might suggest that the gB in EVs produced in MJS cells has modifications and/or a conformation better recognized by the serum. It should be indicated that “empty” EVs could also bind low amounts of anti-BoHV-1 serum, beyond the level of the BSA control. One possible explanation might be that certain structures in EVs have immunoglobulin-binding properties. In addition to antibody binding by gB-positive vesicles from stable cell lines, the serum reacted specifically (when compared to the reaction with “empty” fractions ØF11/ØF21) with EVs-enriched fractions, vF7, vF11, vF15, and the virus-enriched fraction v21, isolated from BoHV-1-infected cell supernatants. The ELISA signal obtained from the bottom fraction of OptiPrep gradients was the strongest, and the BoHV-1-infected MDBK material reacted more efficiently than the MJS-derived antigens, most probably reflecting the level of infection.

## 4. Discussion

Glycoprotein B is a critical factor in herpesvirus infections. Since numerous reports have described gB as the virus entry mediator for example, [52,53,54,55,56], the main goal of our study was to provide new data on the roles of alphaherpesvirus gB expressed in the cells, which should correspond to the situation in of the post-entry phases of infection, i.e., when it appears as a product of an early–late gene [57,58,59]. We could address its localization in the endosomal and lysosomal compartments, including CD63- and MHC II-positive structures, its targeting to EVs and its immunomodulatory effect on the cellular MHC II. A high conservation of the sequence and structure between alphaherpesvirus gB homologs [24] would suggest the preservation of function. However, some differences in the properties of alphaherpesvirus gB homologs have been reported before, like the differential affinity towards the paired immunoglobulin-like type 2 receptors (PILR). HSV-1 gB can bind PILRα molecules, serving as co-receptors of entry [60], whereas PRV gB favors PILRβ [61]. Our results highlight some conserved and also differential features of HSV-1, BoHV-1, and PRV gB. First of all, BoHV-1 and PRV gB shared the ability of their HSV-1 counterpart to localize in extracellular vesicles (Figure 2). To our knowledge, this is the first report on such a property of BoHV-1 and PRV-encoded gBs. Our data, together with prior studies on HSV-1 and HCMV gB [1,4,17,19], strongly suggest that EVs targeting the glycoprotein may be conserved for all herpesviruses. We found EVs incorporation of gB especially notable in stable cell culture supernatants, in the absence of other viral components, and it was rather uncommon for it to depend on the MHC class II status of the cells. While HSV-1 gB was less prominent in MHC II-negative Jurkat-originating EVs (Figure 2D), the T lymphoma cells required activation for an efficient expression from the retrovirus promoter, and the amounts of gB in their EVs possibly reflect the level of expression. Previously, Niazy et al. could isolate gB-positive vesicles from MHC II-negative human fetal IMR90S fibroblasts and COS-7 and, in the present study, PRV gB was enriched in EVs from SK6 cells. Besides, we could also detect gB in EVs from two other MHC II-negative cell lines that we additionally constructed (PRV gB in swine testis ST cell line and BoHV-1 gB in immortalized bovine BoMac macrophages).

A question remains as to what signals drive gB incorporation in EVs. Niazy et al. indicated the conserved in herpesviruses membrane-proximal tyrosine-based YXXф motif (ф is a hydrophobic aa residue), _849_YMAL_852_ in HSV-1 gB, as crucial for EVs targeting. However, this sequence seems to act indirectly as an ER-Golgi release signal, as previously shown for HSV-1, HCMV, and VZV gB [62,63]. The exosomal sorting of proteins has been intensively studied, and the reports show ESCRT-dependent and independent mechanisms, such as engaging tetraspanins, sphingolipids, and ceramide (reviewed in [11]). We found a profound association with CD63, the late endosome-present tetraspanin, to be another conserved feature of all three gB homologs, which may be linked to their exosomal sorting (Figure 3). This is consistent with the previous report by Temme et al., demonstrating the co-localization of CD63 and HSV-1 gB. CD63 has been already highlighted as a critical player in endosomal association and exosomal incorporation of the Epstein–Barr virus (EBV)-encoded oncoprotein latent membrane protein 1 (LMP1) [64,65], another example of a well-characterized EVs-targeted herpesvirus protein. Generally, CD63 has a major role in sorting cargo to EVs, which can be involved in both ESCRT-dependent and independent mechanisms. The CRISPR/Cas9 knockout of CD63 results in a reduced secretion of vesicles [66]. Therefore, further studies are recommended to investigate CD63-herpesvirus–gB interactions, e.g., the incorporation to EVs in CD63 knockout cells. In gB-expressing MJS cells, we did not observe significant alterations in the surface levels of CD63, which indicates its undisturbed trafficking to the plasma membrane. It would be interesting to test whether gB can, like LMP1, form complexes with CD63. Furthermore, the so-called late (L)-domains are involved in binding the ESCRT family protein Alix and CD63, resulting in ILV formation [67]. Interestingly, alphaherpesvirus gB contains a conserved and still unexplored YP(L/I/V)X4L motif in the C-terminal domain, which resembles cryptic (non-canonical) L-domain motifs YPx(n)L/I from Ebola and Marburg virus-encoded VP40 proteins, known for their EVs localization [68]. What is also interesting is that CD63 can regulate EVs and HSV-1 secretion during virus infection [4,69].

BoHV-1 gB and PRV gB appeared as good models to study EVs targeting, as their post-translational modifications, such as N-glycosylation and, in particular, their trans-Golgi-associated cleavage by furin, allowed us to address the gB maturation status in EVs (Figure 2A–C, and Appendix A). As we could detect cleaved gB in EVs with acquired endoH-resistant complex N-glycans, we could conclude that gB can mature and pass the secretory pathway at least to TGN, before EVs incorporation. Our data are in line with a previous report on HSV-1 gB, which has been suggested to end up in late endosomes and EVs directly from the secretory pathway by unknown mechanisms and bypassing endocytosis [23].

One of the most significant, in our opinion, findings of this study was the presence of all three gBs in the EVs-enriched fractions produced during virus infection (Figure 7). Their amounts were small, which may have resulted from the interactions of gB with other viral proteins and its preferential incorporation in virions. We also deliberately maintained the infection conditions at a low level to allow for EVs accumulation and avoid a massive contamination with the lysed cellular material. Another possible explanation could be that gB traffics to EVs only in a particular time window during infection—for example, before interacting viral proteins appear in more significant amounts or virus-induced changes in cellular morphology occur. gB is one of the most immunogenic herpesvirus proteins, raising virus-neutralizing antibodies [70,71,72]. gB-positive EVs from stable MJS and MDBK cells could efficiently bind BoHV-1-raised serum antibodies in ELISA (Figure 8), and EVs-enriched fractions from virus-infected cells also had this potential. Besides, we also detected other highly immunogenic proteins, like gD, in EVs from infected cells (Figure 7). Their incorporation in EVs in the case of HSV-1 was not previously addressed by Deschamps and Kalamvoki [4]. One can imagine that if such antibody binding by EVs can also occur in in vivo conditions, the vesicles would have the potential to affect the serum level of neutralizing antibodies or cellular responses. The idea that secreted vesicles bearing viral proteins could act as decoys that might trap antiviral antibodies, reducing detection of infectious virions, has recently gained more recognition (e.g., [73]). Such a potential has been reported for hepatitis B virus surface antigen-carrying microvesicles (known as subviral particles [74]) or Ebola virus glycoprotein-decorated pseudoparticles [75]. This way viruses would share another immunomodulatory strategy with cancer cells as tumor exosomes can sequester tumor-reactive antibodies and reduce antibody-dependent cellular cytotoxicity [76]. Another type of a decoy strategy is employed by human immunodeficiency virus (HIV), and it requires interaction of HIV particles and exosomes from infected cells to facilitate viral transfer to uninfected cells [77]. It is a subject that needs to be further investigated for alphaherpesvirus gB. HSV-1 EVs have been already reported to mediate innate immunity by transferring the stimulator of interferon genes (STING) and viral microRNA miR-H28 to uninfected cells, sensitizing the recipient cells to forthcoming infection [4,78]. It would be interesting to check if during BoHV-1 or PRV infection, EVs can also shape the immune response.

Another important post-entry function of gB is MHC II downregulation and the boosting of their loading to EVs, as reported for the HSV-1 glycoprotein by a series of elegant studies from the Koch/Eis-Hübinger group [19,23,79,80]. According to our data, all three gB homologs co-localized equally well with MHC II and CD63, increasing the MHC II-CD63 overlap parameters (Figure 3). CD63 was previously shown to interact with HLA-DR at MVB and regulate the DR expression in MJS cells [46]. Thus, the differential effect of the three gB homologs came as a surprise, with the HSV-1 glycoprotein acting upon MHC II (DR haplotype) with the highest potency, BoHV-1 gB with an intermediate activity, and PRV gB as the most divergent. Furthermore, we report that this differential effect was not restricted to human MHC II, as the same tendency could be observed in bovine cells. In addition, for HSV-1 and BoHV-1 gB, we could demonstrate, by internalization/export assays, that gB expression contributed to the retardation of HLA-DR on the way to the cell surface, which most probably accounts for their lower plasma membrane levels (Figure 5). It should be noted that BoHV-1 gB could affect MHC II transport to an even greater extent than its HSV-1 counterpart. These results are in agreement with the conclusions from the immunofluoresce analysis by Niazy et al. on HLA-DR trapping in early endosomes by HSV-1 gB. One possible explanation can be that the effect of gB on DR complexes, indeed, predominantly depends on the Ii-like motif from the N-terminal domain and involves direct binding, as presented for HSV-1 gB in [79]. Then, the low sequence conservation within this particular N-terminal fragment of gB could explain the variability between the homologs (Figure 5A). Dependence on the Ii-like motif would be in line with the observed reduced complex formation by BoHV-1 gB and MHC II (Figure 5C) and most probably with the lack of PRV gB affinity towards HLA-DR (Figure 5D). On the other hand, the BoHV-1 gB sequence is, in this respect, already so divergent, with the DR-binding lysine residues absent, that we can also speculate that this region is an essential structural element of gB involved in interactions. Unfortunately, although the post-fusion structure of the PRV and HSV-1 gB ectodomain has been resolved [24,55,81], the visualized HSV-1 gB construct lacks the region with the Ii-like motif for comparison.

The differential effect of gB on the MHC II surface expression seemed to be associated with the adverse levels of MHC II in EVs in the presence of PRV gB. This observation indicates that the mechanism of MHC II regulation is unlikely to be based only on binding MHC II and bridging them to the EVs-loading machinery. It may, instead, involve more general perturbations of the vesicular transport or endosomal architecture within a cell. In our stable cell lines, in general, we could hardly and only for HSV-1 gB demonstrate alterations to endosomes marked by LAMP-1 and Rab7 (Appendix A), which, most probably, as suggested by Temme et al., coincided with moderate levels of expression. It would be interesting then to investigate whether HSV-1 gB, BoHV-1, and PRV gB differ in the activity of inducing the homotypic fusion of early endosomes. PRV gB should have this potential, as demonstrated in [23] (for the gB of the Prophylaxia strain). Finally, from the immunological point of view, an interesting result of this study was the enrichment in HLA-DR in the EVs from all three alphaherpesvirus-infected cells. Despite the differential activity of PRV gB in stable cell lines and the low level of infection in MJS cells, the PRV effect on the DR content in EVs was potent. We may expect that, during virus infection, we can observe the cumulative effects of many viral components. For HSV-1, besides gB, gene products UL41 (Vhs), γ34.5, and US1 have been shown to target the MHC II pathway [82,83]. Vhs activity towards MHC II was confirmed for BoHV-1 [84], whereas for PRV, the host shutoff is generally weaker and delayed [85]. The increased HLA-DR incorporation to EVs may not be directly dependent on gB. Unfortunately, it would not be easy to assess the role of gB in this respect during infection, as it is an essential gene product, and such an investigation would, for example, require silencing the gB in infected cells.

## 5. Conclusions

Despite the high conservation of the sequence and structure between HSV-1, BoHV-1, and PRV gB (especially between the last two viruses, belonging to the same *Varicellovirus* genus), we report here the properties shared by the three alphaherpesvirus gB homologs, their differential activities in extracellular vesicle targeting and their effect on MHC class II molecules. EVs incorporation seems to be a generally conserved property of herpesvirus gB, which coincides with late endosome marker CD63 co-localization and does not require endogenous MHC II expression. HSV-1, BoHV-1, and PRV gB could also increase CD63-MHC II co-localization in MJS cells. BoHV-1 glycoprotein shared the HSV-1 gB tendency to bind MHC II (human or bovine), cause the retardation of HLA-DR transport to the cell surface, and decrease the surface MHC II levels. In this respect, PRV gB showed an adverse activity, most evident in the increase in the MHC II surface levels and decrease in the HLA-DR loading to EVs. In EVs isolated from virus-infected cells by iodixanol gradient ultracentrifugation, small amounts of gB, but also other highly immunogenic viral glycoproteins, like gD, were detected. BoHV-1 gB-containing EVs and BoHV-1-infected cell-derived EVs could bind whole-virus-immunized animal serum antibodies in ELISA, suggesting their decoy potential. Finally, EVs from HSV-1, BoHV-1, and PRV-infected MJS were enriched in HLA-DR. During their evolution, herpesviruses master immune evasion properties, and each new described immunomodulatory mechanism may be important for the development of anti-herpesvirus vaccines. While the functional consequence of MHC II downregulation or the secretion of MHC II-enriched EVs, especially for the stimulation of CD4+ T cells, both in vitro and in vivo, remains unknown, we find the potential of HSV-1 gB, and also that of BoHV-1 gB, from stable cell lines to be compelling.

## Figures and Tables

**Figure 1 viruses-12-00429-f001:**
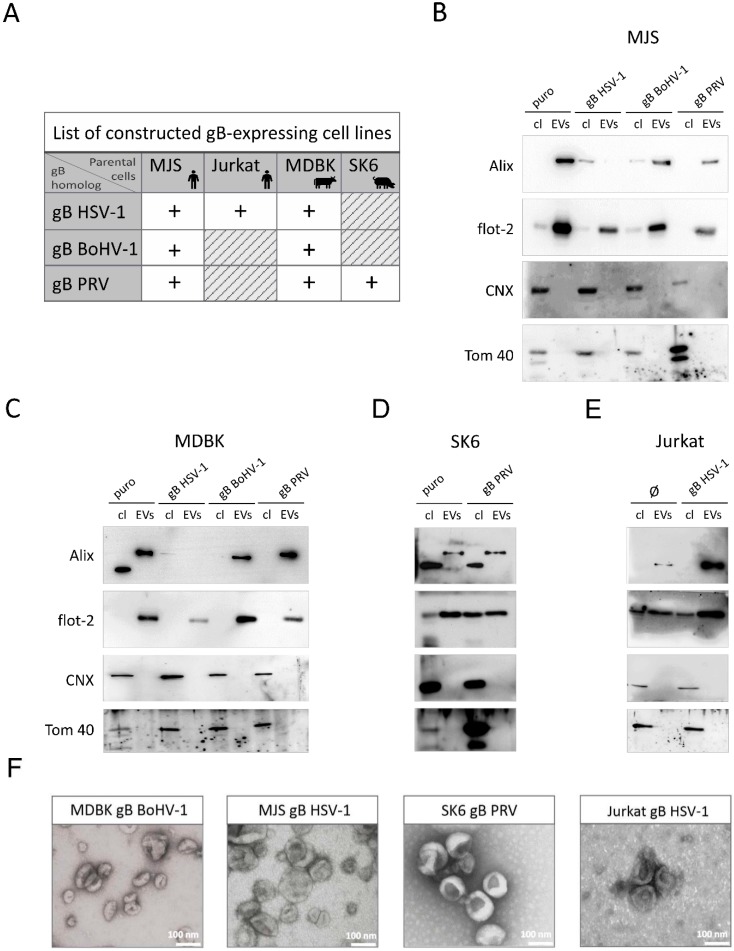
Construction of stable cell lines expressing bovine herpesvirus 1 (BoHV-1), pseudorabies virus (PRV), or herpes simplex 1 (HSV-1) glycoprotein B (gB), and characterization of size exclusion chromatography (SEC)-isolated extracellular vesicles (EVs). (**A**) The list of constructed cell lines expressing individual viral gB homologs. Human melanoma Mel JuSo (MJS) and Madin–Darby bovine kidney (MDBK) produce endogenous major histocompatibility (MHC) class II. The Human Jurkat T cell line and swine kidney SK6 cells are MHC II-negative. (**B**–**E**) Immunoblotting detection of EVs markers Alix and flotillin-2 (flot-2) in cell lysates (cl) of constructed cell lines or SEC-isolated EVs. Calnexin (CNX) and mitochondrial Tom 40 were used as non-EVs control proteins. MJSpuro, MDBKpuro, SK6puro, or untransduced Jurkat cells (Ø) were used as negative controls. (**F**) Representative transmission electron microscopy images of EVs preparations from gB-expressing species-specific cells; scale bar 100 nm.

**Figure 2 viruses-12-00429-f002:**
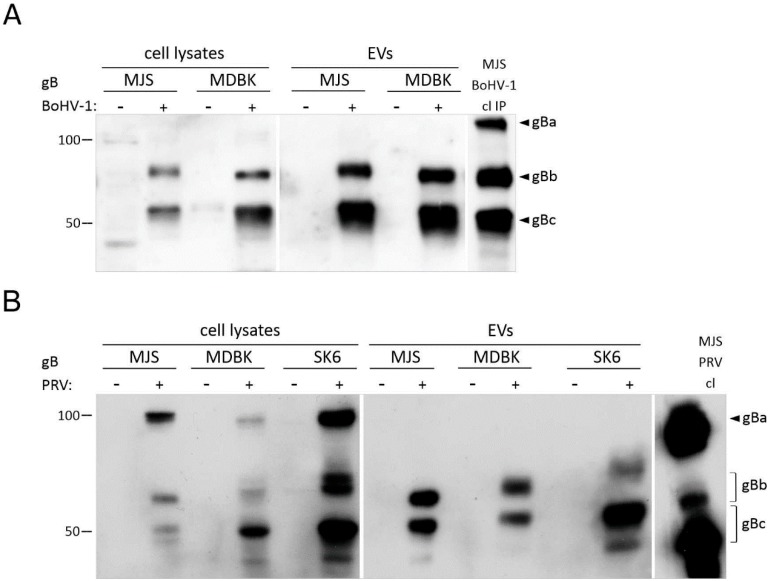
Immunoblotting detection of BoHV-1 (**A**), PRV (**B**), and HSV-1 gB (**C**) in cell lysates (cl) of constructed cell lines or SEC-isolated EVs. (+): gB-expressing cells; (−): MJSpuro, MDBKpuro, SK6puro, or untransduced Jurkat cells (Ø) were used as negative controls. Virus-infected cell lysates (cl) were analyzed as positive controls; in the case of BoHV-1, gB was immunoprecipitated (IP) with H2 antibody before immunoblotting with anti-BoHV-1 serum. Size markers are in kilodaltons. gBa: uncleaved BoHV-1 and PRV gB precursor; gBb: cleaved N-terminal gB subunit; gBc: cleaved C-terminal gB subunit.

**Figure 3 viruses-12-00429-f003:**
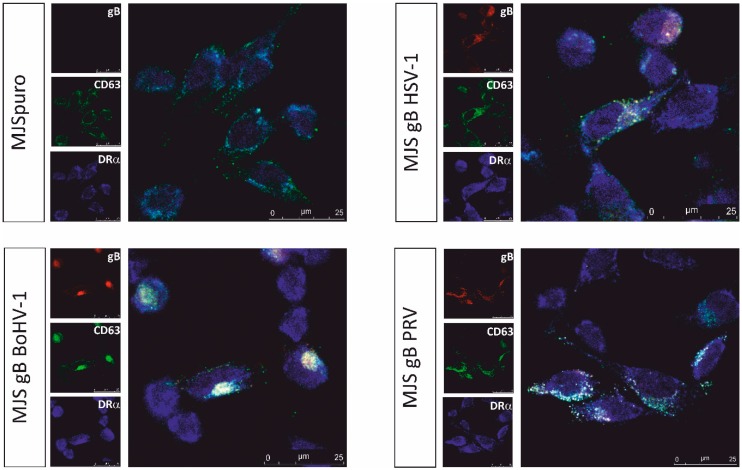
Intersection of gB, late endosome/exosome marker CD63 and major histocompatibility (MHC) class II (DRα) trafficking. Subcellular localization was analyzed by immunofluorescence-confocal laser scanning microscopy. gB was stained with specific mAb and Alexa 546-conjugated anti-mouse IgG (**red**), CD63 with goat antibodies and Alexa 488-conjugated anti-goat IgG (**green**) and the DRα subunit of MHC II with rabbit antibodies and Alexa 633-conjugated anti-rabbit IgG (**blue**).

**Figure 4 viruses-12-00429-f004:**
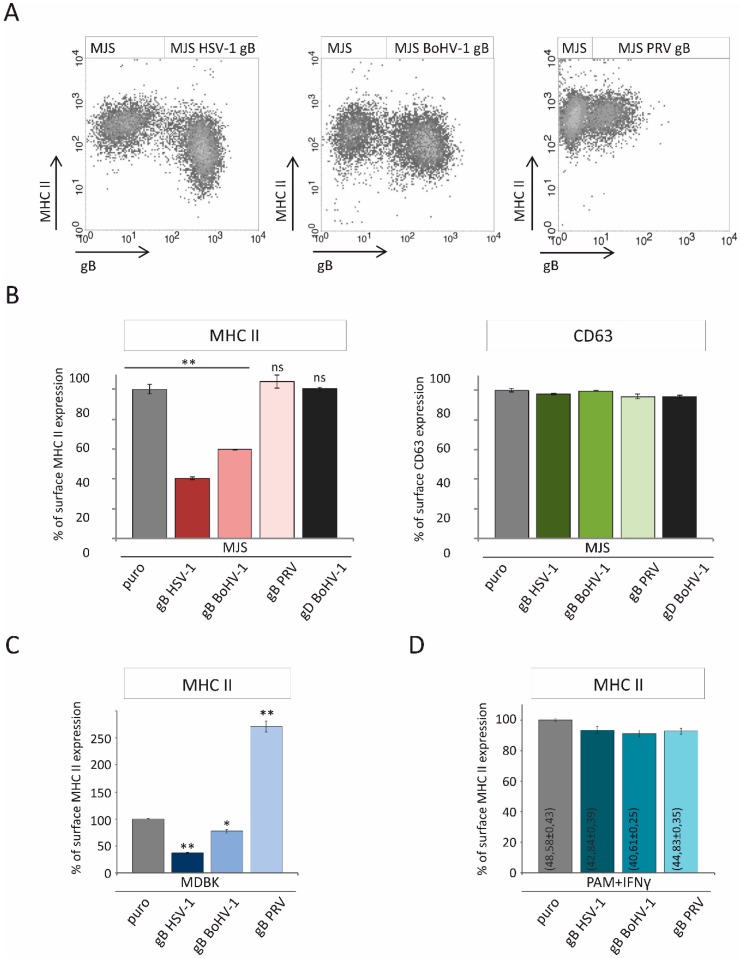
The effect of alphaherpesvirus gB expression on the surface levels of human, bovine, and porcine MHC class II. (**A**) Surface expression of HLA-DR was assessed by flow cytometry on gB-retrovirus transduced unsorted MJS cells (a nerve growth factor receptor (NGFR) marker reflecting gB expression is plotted against MHC II). (**B**) Analysis of the relative HLA-DR levels on the surface of gB-expressing sorted cells. Surface CD63 expression was analyzed as a control. The effect of the BoHV-1-encoded gD on surface MHC II and CD63 was assessed for comparison. The mean fluorescence intensity of MHC II/CD63 on MJSpuro control cells was set at 100%. The analysis was performed in triplicate. The statistical significance of differences between MJSpuro and MJS gB/gD cell lines was estimated by *t*-test; ** *p* ≤ 0.001; ns-not statistically significant. (**C**) Surface expression of bovine MHC II was analyzed by flow cytometry on gB-retrovirus transduced sorted MDBK cells and MDBKpuro control cells. The mean fluorescence intensity of MHC II in MDBKpuro control cells was set as 100%. The analysis was performed in triplicate. The statistical significance of differences between MDBKpuro and MDBK gB cell lines was estimated by *t*-test; ** *p* ≤ 0.001, * *p* ≤ 0.01. (**D**) Surface expression of porcine MHC II was analyzed by flow cytometry on gB-retrovirus transduced sorted porcine alveolar macrophage (PAM) cells and PAMpuro control cells. The cells were incubated for 36 *h* with 0.1 mg mL^−1^ of recombinant swine interferon (IFN)-γ. The mean percentage of MHC II-expressing cells (±standard deviation) is provided on each bar. The mean fluorescence intensity of MHC II in PAMpuro control cells was set as 100%. The analysis was performed in triplicate. The differences between PAMpuro and PAM gB cell lines were found statistically insignificant.

**Figure 5 viruses-12-00429-f005:**
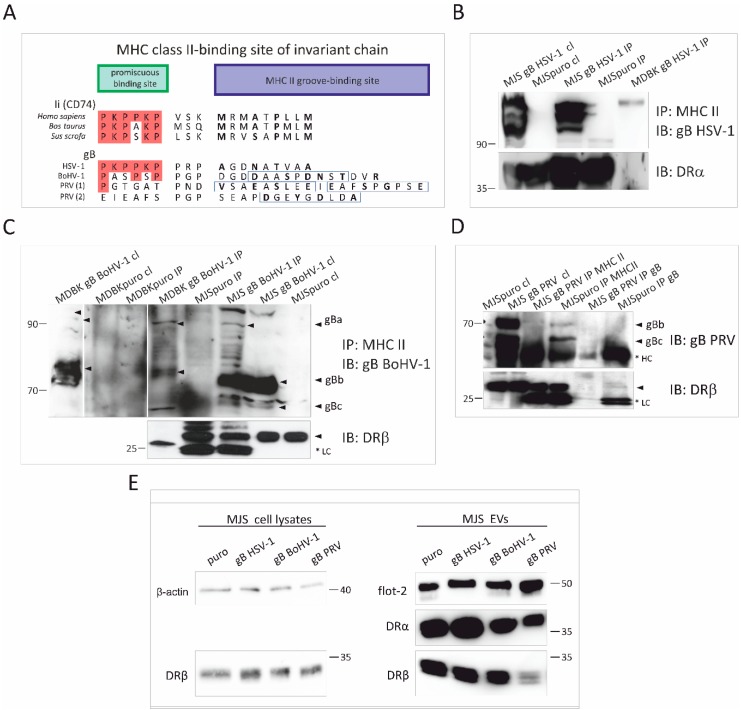
Analysis of interaction between MHC II and HSV-1, BoHV-1 or PRV gB, and the effect of alphaherpesvirus gB on MHC II incorporation in EVs. (**A**) Comparison of amino acid sequences representing potential HSV-1, BoHV-1, and PRV gB and MHC II biding site. HSV-1 gB contains an allotypic HLA-DR-binding region homologous to the human invariant chain (Ii, CD74), consisting of the promiscuous and the MHC II groove-binding sites. Whereas bovine and porcine Ii sequences are highly conserved, their corresponding regions in BoHV-1 and PRV gB (two candidate sequences) are variable. Amino acids (aa) that are identical to those in human Ii are in red frames. Boxes highlight predicted MHC II peptides. The groove-binding positions are in bold. (**B**) Complex formation between HSV-1 gB and HLA-DR or bovine MHC II from cell lysates was evaluated by co-immunoprecipitation (IP) using anti-MHC II mAb. Immunoprecipitated proteins were separated by SDS-PAGE and detected by immunoblotting (IB) with anti-gB or anti-DRα antibodies (for human cells). (**C**) and (**D**) Co-IP with anti HLA-DR or bovine MHC II (in C) mAb from MJS BoHV-1/PRV gB or MDBK BoHV-1 gB lysates. gB or DRβ were detected in SDS-PAGE-separated complexes by IB. Size markers are in kilodaltons. *LC: light chain of antibodies; HC: heavy chain of antibodies. Arrows indicate gB species. (**E**) Detection of DRα, DRβ and flotillin-2 (flot-2) as an EVs marker in SEC-purified EVs by IB. β-actin and DRβ were detected in cell lysates for comparison. Size markers are in kilodaltons.

**Figure 6 viruses-12-00429-f006:**
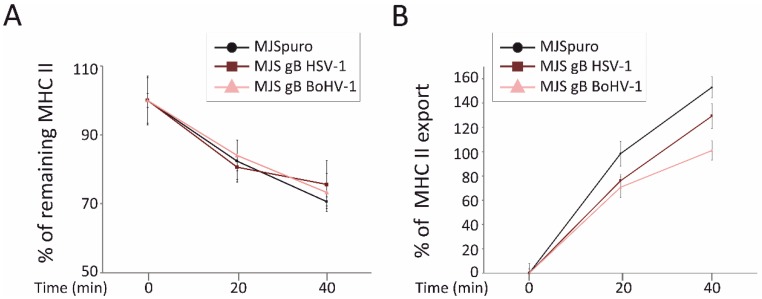
HSV-1 and BoHV-1 gB do not affect internalization but contribute to a retarded appearance of MHC II at the cell surface. (**A**) An internalization assay was performed on MJS HSV-1 gB (dark brown line), MJS BoHV-1 gB (pink line), or control MJSpuro (black line) cells stained on ice with anti-MHC class II L243 antibody and PE-conjugated IgG. The cells were then shifted to 37 °C for 20 or 40 min. The mean fluorescence intensities of remaining MHC II were assessed by flow cytometry for triplicate samples and compared with MHC II at time point 0 (set as 100%). The differences between MHC II levels were statistically insignificant. (**B**) For the MHC II export assay, the cells were stained on ice with a saturating amount of anti-MHC class II L243 antibody and incubated at 37 °C for 20 or 40 min in the presence of allophycocyanin (APC)-conjugated L243 MAb detecting newly arrived MHC II. The mean fluorescence intensities of MHC II staining were normalized to the values for time point 0 samples and depicted as a percentage of the increase in MHC II appearance on MJSpuro (black line) or MJS-gB cells. Results are representative of three independent experiments. The statistical significance of differences between MHC II on MJSpuro and gB-expressing cells at 20 or 40 min was estimated by a *t*-test; *p* ≤ 0.01.

**Figure 7 viruses-12-00429-f007:**
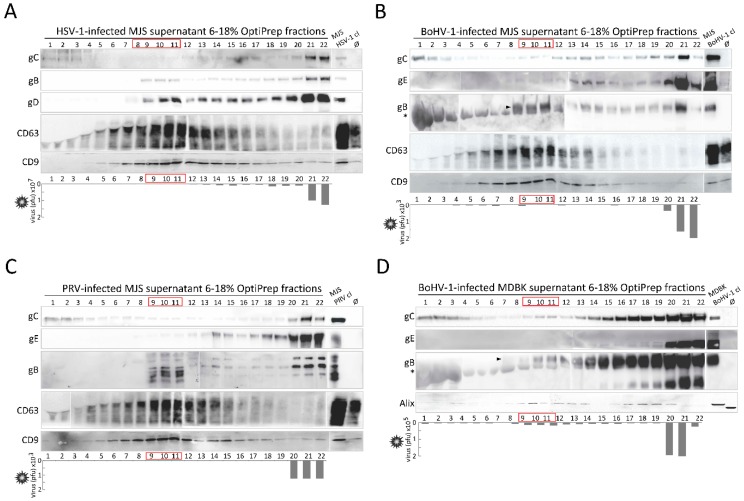
Alphaherpesvirus gB localizes to EVs-enriched fractions isolated from virus-infected MJS cells and BoHV-1-infected MDBK. OptiPrep gradient-purification of EVs-enriched and virion-enriched fractions from HSV-1-infected MJS (**A**), BoHV-1-infected MJS (**B**), PRV-infected MJS (**C**), and BoHV-1-infected MDBK (**D**) supernatants. MJS cells were infected with HSV-1 at a multiplicity of infection (moi) of 0.1, with BoHV-1/PRV at an moi of 0.5; MDBK cells were infected at an moi of 0.1. After 50 h post-infection, 0.5 mL fractions were collected from 6%–18% iodixanol (Optiprep) gradient by ultracentrifugation. The 40 µL samples were analyzed in denaturing (for viral glycoprotein C-gC, glycoprotein B-gB, glycoprotein D-gD, for HSV-1, glycoprotein E-gE, for BoHV-1 and PRV or EVs marker Alix) or non-denaturing (for EVs markers CD63 and CD9) SDS-PAGE, and immunoblotted with specific mAbs or polyclonal anti-PRV gB serum. Uninfected (Ø) or virus-infected cell lysates (cl) were analyzed as controls. Bottom panels represent virus titres in each fraction measured by plaque assay. (**E**) Proteins from the indicated fractions (40 μL) were separated by SDS-PAGE, and immunoblotted with antibodies specific to the ICP5/VCP5 homolog of HSV-1 and BoHV-1, virulence factor γ34.5 from HSV-1, immunomodulatory UL49.5 protein of BoHV-1 and envelope glycoproteins gD, and gM, as indicated. HLA-DRβ was detected in the fractions as an EVs marker. Cell lysates (cl) and virus-infected cell lysates (30 μg) were analyzed as controls. (**F**) Comparison of HLA-DRβ levels between EVs from virus-infected and uninfected MJS. DRβ and flotillin-2 (flot-2, EVs marker) from F11 fractions were immunoblotted with specific mAb. Numbers indicate the optical density of DRβ bands from virus-infected MJS EVs, normalized to flot-2 from the same samples, relative to DRβ in EVs from uninfected MJS (set as 1).

**Figure 8 viruses-12-00429-f008:**
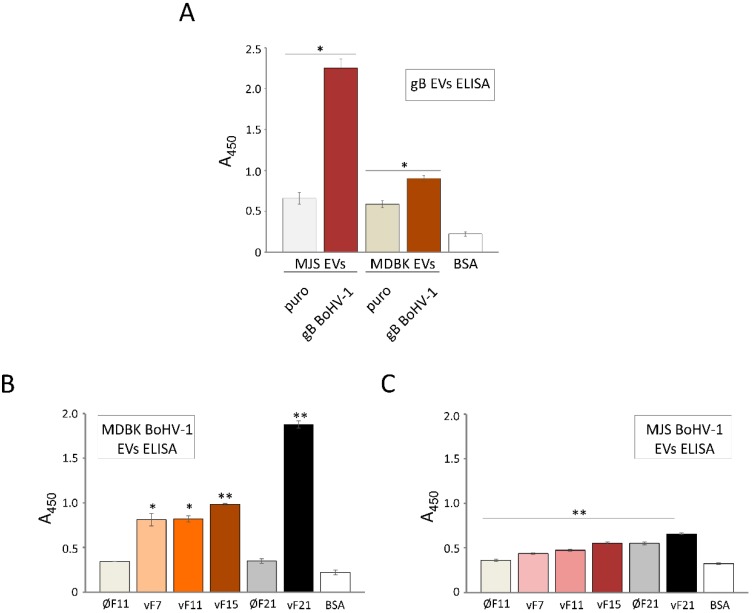
EVs-transferred BoHV-1 proteins or specifically gB can bind virus-specific antibodies from the whole virus-specific animal serum. (**A**) EVs gB ELISA was performed with the goat anti-BoHV-1 serum.. The signal from gB-containing EVs was compared with the signal from MJS/MDBKpuro. (**B**) and (**C**) ELISA detection of BoHV-1 proteins in OptiPrep-isolated material (fractions vF11, vF15 and vF21) from BoHV-1-infected MDBK (**B**) or MJS (**C**) cells. The signal from virus-infected material was compared with the signal from the corresponding fractions F11 or F21 from uninfected MJS/MDBK (designated as Ø). A total of 1% BSA in PBS served as a background control. Each reaction was performed in triplicate and bars represent means of absorbance at 450 nm with the standard deviations; * *p* ≤ 0.01, ** *p* ≤ 0.001.

**Table 1 viruses-12-00429-t001:** Analysis of gB/CD63/MHC II (DRα) co-localization by Pearson’s correlation coefficient measurement using the Leica Appilication Suite X. The data correspond to mean ± SD of at least five independent pictures; not determined (nd).

Cell Line	Compared Channels	Pearson’s Correlation ± SD
**MJSpuro**	gB-CD63	nd
	gB-DRα	nd
	**CD63-DRα**	**0.64 ± 0.02**
**MJS gB HSV-1**	gB-CD63	0.84 ± 0.02
	gB-DRα	0.64 ± 0.04
	**CD63-DRα**	**0.71 ± 0.07**
**MJS gB BoHV-1**	gB-CD63	0.92 ± 0.01
	gB-DRα	0.68 ± 0.08
	**CD63-DRα**	**0.76 ± 0.07**
**MJS gB PRV**	gB-CD63	0.94 ± 0.01
	gB-DRα	0.70 ± 0.02
	**CD63-DRα**	**0.74 ± 0.02**

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
