# Peer review of "Alphaherpesvirus gB Homologs Are Targeted to Extracellular Vesicles, but They Differentially Affect MHC Class II Molecules"

_viruses, 2020, doi:10.3390/v12040429_

Round 1

Reviewer 1 Report

This is an interesting study on a potential immunoevasion strategy used by herpesvirus, and in particular glycoprotein B (and homologs) interacting with the MHC II pathway.

I find the study to be well performed and written and has novel findings. I have some minor suggestions:

  1. I would encourage moving Figure S1 into the main paper. Characterisation of the EVs is critical to this type of study and the evidence (which is good in this manuscript) I think should always be presented in the main body.
  2. I would discourage the use of the term 'gold-standard' (line 353 of pdf). Although SEC is indeed very useful, there is still much contention in the EV field as to the best technique for isolation. It is likely dependent on several factors, source material, end-use etc, so I would suggest use of some other term that will prevent the work from becoming non-gold standard if the field moves on to other technologies.
  3. in section 3.1.2, lines 373 to 395 in the pdf, I cannot detect any Figure being referred to ( I presume it is Fig 1). Please amend as necessary.
  4. Latterly, the ELISA data is used to argue for a 'decoy' evasions strategy. I think this needs supporting text to expand the hypothesis more, and cite any examples of similar evasion straggles of this sort.

Author Response

Response to Reviewer 1 Comments

Point 1: I would encourage moving Figure S1 into the main paper. Characterisation of the EVs is critical to this type of study and the evidence (which is good in this manuscript) I think should always be presented in the main body.

Response 1: Thank you very much for this comment. We were debating on including this Figure in the main manuscript before, and we agree it is important for characterization of our preparations. Therefore, we moved Figure S1 to Figure 1, now as 1B. At the same time the previous Figure 1C-E was moved as the new Figure 2, which also changed Figure numbering in the ms.

Point 2: I would discourage the use of the term 'gold-standard' (line 353 of pdf). Although SEC is indeed very useful, there is still much contention in the EV field as to the best technique for isolation. It is likely dependent on several factors, source material, end-use etc, so I would suggest use of some other term that will prevent the work from becoming non-gold standard if the field moves on to other technologies.

Response 2: We fully agree with this comment, and we replaced “gold-standard” by “widely used”, now in line 302.

Point 3: in section 3.1.2, lines 373 to 395 in the pdf, I cannot detect any Figure being referred to ( I presume it is Fig 1). Please amend as necessary.

Response 3: We apologize for this missing reference. We included references to the new Figure 2 in lines 388 and 394.

Latterly, the ELISA data is used to argue for a 'decoy' evasions strategy. I think this needs supporting text to expand the hypothesis more, and cite any examples of similar evasion straggles of this sort.

Response 4. We expanded the discussion, lines 964-973, as follows: The idea that secreted vesicles bearing viral proteins could act as decoys that might trap antiviral antibodies, reducing detection of infectious virions has recently gained more recognition (e.g., [72]). Such a potential has been reported for hepatitis B virus surface antigen-carrying microvesicles (known as subviral particles [73]) or Ebola virus glycoprotein-decorated pseudoparticles [74]. This way viruses would share another immunomodulatory strategy with cancer cells as tumor exosomes can sequester tumor-reactive antibodies and reduce antibody-dependent cellular cytotoxicity [75]. Another type of a decoy strategy is employed by human immunodeficiency virus (HIV), and it requires interaction of HIV particles and exosomes from infected cells to facilitate viral transfer to uninfected cells. It is a subject that needs to be further investigated, also for alphaherpesvirus gB

Reviewer 2 Report

In this manuscript the authors studied the incorporation of bovine herpesvirus-1 gB and pseudorabies virus gB into extracellular vesicles (EVs) and compared them to the level of incorporation of herpes simplex-1 (HSV-1) gB. They tested EVs collected from stable cell lines expressing gB as well as virus infected cells. In addition, the authors check co-localisation of gB with EVs markers like CD63 and CD9 and the effect on MHC class II molecules expression.

Not very surprising, the authors found that BoHV-1 gB and PRV gB can be found in EVs secreted from stable cell lines expressing those gBs. As also noted by the authors, gB is one of the best documented EVs incorporated viral proteins. Taken the high conservation of gB within the herpesvirus family it was rather expected that BoHV-1 gB and PRV gB can also be found in EVs.

The authors further separated populations of EVs from infected cells. The authors claimed these vesicles contain gB (and other herpesvirus proteins) based on WB of fractions from density gradient. I find these results not very convincing, to start with the authors should provide blots for all fractions as they did for CD63 and gC (Supp Fig 5), why only specific fractions are provided?

To support their claim, the authors also provided negative stain images of some vesicles. The problem with negative stain images is that they can be very selective. To support their claim that gB is indeed incorporated into the EVs the authors should either do immunogold labelling or cryoEM, as gB can very easily be identified on EVs (see PNAS 2016 113:4176). 

Overall, this manuscript don’t show much originality and don’t provide novel data. The authors set their main goal ‘to contribute new data on the roles of alphaherpesvirus gB in the post-entry phases of infection’ but they do not deliver on that. The data on co-localisation with MHC II and CD63 is anecdotal and its relevant for the field is not clear.

Unrelated, this manuscript requires major editing as in its current form is very hard to read and follow.
